# State-of-the-Art Review on the Analytic Hierarchy Process with Benefits, Opportunities, Costs, and Risks

Antonella Petrillo [1], Valerio Antonio Pamplona Salomon [2,*] and Claudemir Leif Tramarico [2]

1  Department of Engineering, University of Naples "Parthenope", 80143 Napoli, Italy; antonella.petrillo@uniparthenope.it
2  Department of Production, Universidade Estadual Paulista (UNESP–Sao Paulo State University), Guaratingueta 12516-410, SP, Brazil; claudemir.tramarico@unesp.br
*  Correspondence: valerio.salomon@unesp.br

**Abstract:** The benefits, opportunities, costs, and risks (BOCR) model is a multiple-criteria decision-making (MCDM) model used to elicit a mutually exclusive and collectively exhaustive set of criteria. As an acronym proposed in the theory of the analytic hierarchy process (AHP), the BOCR model has received attention from users of this MCDM method. A state-of-the-art review, an approach to a literature review that is more comprehensive than a rapid review but not as exhaustive as a systematic literature review, was performed with the Scopus database. The overwhelming majority of documents found on BOCR were practical applications, but they were from diverse areas, including business, computer science, and engineering. It is proposed that two main kinds of contributions for future research on BOCR should be methodological and practical.

**Keywords:** AHP; BOCR; literature review; MCDM

## 1. Introduction

Multiple-criteria decision making (MCDM) is "the study of methods and procedures by which concerns about multiple conflicting criteria can be formally incorporated into the management planning process" (International Society on Multiple Criteria Decision Making 2022). Moreover, MCDM is a methodology, a collection of methods developed from the 1960s to solve decision problems (Greco et al. 2016). Most MCDM methods are known by their acronyms; we can list a few alphabetically (Saaty and Ergu 2015):

- Additive Ratio Assessment(ARAS);
- Analytic Hierarchy Process (AHP);
- Analytic Network Process (ANP);
- Complex Proportional Assessment (COPRAS);
- Decision-Making Trial and Evaluation Laboratory (DEMATEL);
- Elimination et Choix Traduisant la Realite (ELECTRE);
- Goal Programming (GP);
- Measuring Attractiveness by a Categorical-based Evaluation Technique (MACBETH);
- Multi-Attribute Utility Theory (MAUT);
- Preference Ranking Organization Method for Enrichment of Evaluations (PROMETHEE);
- Technique for Order Preference by Similarity to Ideal Solution (TOPSIS);
- Visekriterijumska Optimizacija i Kompromisno Resenje (VIKOR).

As there are so many MCDM methods, classifications have arisen. One of the first classifications was the American School versus the European School (Vincke 1992). That classification was criticized, not only for xenophobia but also in terms of difficult developments by international teams (Olson 1996). Aggregation approaches and outranking approaches are better classifications (Ishizaka and Nemery 2013). However, both sets of approaches often share the same result (Salomon and Rangel 2015). For instance, the AHP

and MAUT are MCDM methods with an aggregation approach, and they are from the American School; ELECTRE and PROMETHEE are MCDM methods with an outranking approach from the European School. MACBETH, an MCDM method with an aggregation approach from the European School, is an exception.

The choice of an MCDM method should be based on characteristics of the decision problem (Ishizaka and Nemery 2013). Nevertheless, the choice of an MCDM method has been treated as a matter of opinion (Salomon et al. 2013). This paper presents a review of the AHP, a leading MCDM method for decades (Khan et al. 2018; Tramarico et al. 2015; Wallenius et al. 2008). One main reason for the AHP's leadership in MCDM is its solid mathematical foundation. Criticisms of AHP practice and theory, including those related to ranking reversal (Belton and Gear 1983), inconsistency judgments (e Costa and Vansnick 2008), and the Saaty Scale (Salo and Hamalainen 1997), have all been refuted (Garuti Anderlini et al. 2008; Saaty et al. 2009).

AHP applications may be facilitated by using software packages developed by consultancy companies (Creative Decisions Foudation 2023; Expert Choice, Inc. 2022). There are also packages developed for www.matlab.com (accessed on 21 April 2023) and www.r-project.org www.matlab.com (accessed on 21 April 2023). Moreover, the steps for AHP and ANP applications are disclosed in several books and journals (De Felice et al. 2016; Saaty 1977, 1980; Salomon 2016). Therefore, both AHP and ANP can be easily applied with spreadsheets. Other MCDM methods require proprietary software, such as M-MACBETH (Salomon 2008).

Due to the disclosure of the steps for AHP, which include the employment of hierarchies, pairwise comparison matrices, eigenvalues, and eigenvector estimation, AHP theory has been developed through practical contributions. One major example is the analysis of benefits, opportunities, costs, and risks (BOCR). The use of an MCDM model with these four main criteria aims for a mutually exclusive and collectively exhaustive (MECE) set of criteria (Lawrence and Pasternack 2002). Mutually exclusive means that the same criterion is not used twice or more. Collectively exhaustive means that all the important criteria are considered. Developed by the McKinsey consultancy company, the MECE principle is helpful to avoid redundancy among criteria (Chevalier 2016).

An older BOCR reference in the Scopus database is an ANP application document relating to the Iranian wood industry (Azizi et al. 2005). That document refers to a book and a conference paper from Prof. Saaty as seminal works on BOCR (Saaty 1997). Google Scholar highlights Wijnmalen (2007), an article on BOCR's methodology cited by 118 other works in the Scopus database.

This paper intends to answer the question: how have the AHP and BOCR been applied? Therefore, the main objective of this paper is to present the state-of-the-art of the literature on the AHP and BOCR. Specific objectives are to identify research gaps and propose future research themes. This paper is divided into five more sections. Section 2 presents the background to BOCR, Section 3 discusses the research methodology, Section 4 presents the results, Section 5 includes a discussion and a summary of the results, and Section 6 offers the conclusions and suggestions for future work.

## 2. Background

Despite there being several methods for MCDM, their application follows the same three major steps: structuring, measuring, and synthesis (Salgado et al. 2012). Structuring consists of the identification of the elements of the decision problem: alternatives, criteria, and the objective. Measuring is the step in which weights are attributed to the criteria and also to the alternatives according to each criterion. Finally, in the synthesis, the weights are combined, resulting in overall weights for each alternative. The way in which each step is implemented is what differentiates MCDM methods from each other. Since the methods were developed by different authors in different countries, there are also differences in terminology (Greco et al. 2016; Ishizaka and Nemery 2013; Triantaphyllou 2000).

The BOCR model is a model for the structuring step allowing investigation of the positives (benefits) and negatives (costs), as in benefit–cost analysis (BCA) approaches (Saaty 1994; Wijnmalen 2007). As further discussed in this section, the use of a BOCR model also facilitates measuring and synthesis, the other two MCDM steps.

In BCA, one asks if the benefits justify the costs. In the theory of the AHP, BCA was enhanced with the BOCR model, allowing two more merits: opportunities and risks (Saaty 2013). Then, the BOCR merits can be incorporated in the traditional benefit-to-cost ratio analysis used in economics to obtain a ratio outcome for each alternative. In addition to being more complete than BCA, the BOCR model facilitates the application of MCDM by answering one of the first questions that must be asked: what criteria should be considered?

Figure 1 presents a generic three-level BOCR model with a complete hierarchy for a decision problem with three alternatives.

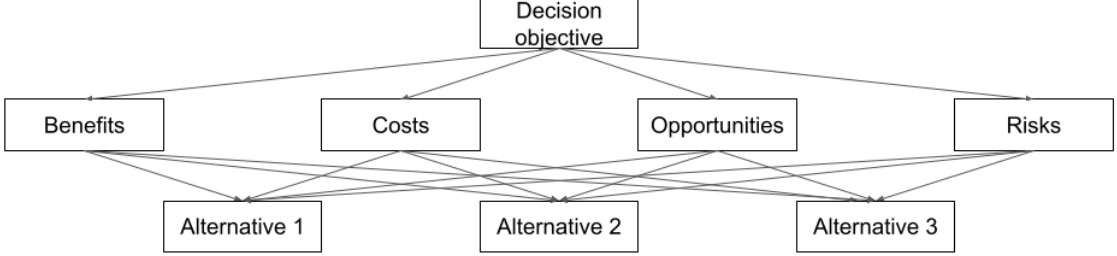

**Figure 1.** Generic three-level BOCR model with a complete hierarchy.

The hierarchy presented in Figure 1 is complete because all the BOCR merits are included, and all alternatives collaborate for each BOCR merit. In contrast, an incomplete hierarchy would be a structure in which not all the elements from a level connect to all elements from the higher level (Carmo et al. 2013). However, sometimes, all BOCR elements may not be present in real-world situations. In some problems, opportunities maybe not be involved, reducing the BOCR model to a BCR model, as in the example of treatment for a heart problem (Saaty 1994). In this problem, alternatives, such as "angioplasty", "bypass operation", "medicinal treatment", "transplant", and "nothing", were evaluated according to three benefit sub-criteria (longer life, mental health, and physical health), four cost sub-criteria (economic costs, mental stress, physical drain, and treatment complications), and three risk sub-criteria (no change, treatment complications, and treatment failure). Obviously, choosing among the alternatives depends on each patient's condition, including age, blood pressure, diabetes, smoking, and weight.

Another kind of incomplete hierarchy occurs when one or more alternatives are not evaluated according to all the criteria. In Figure 2, alternative two is not evaluated according to opportunities, and alternative three is not evaluated according to risks.

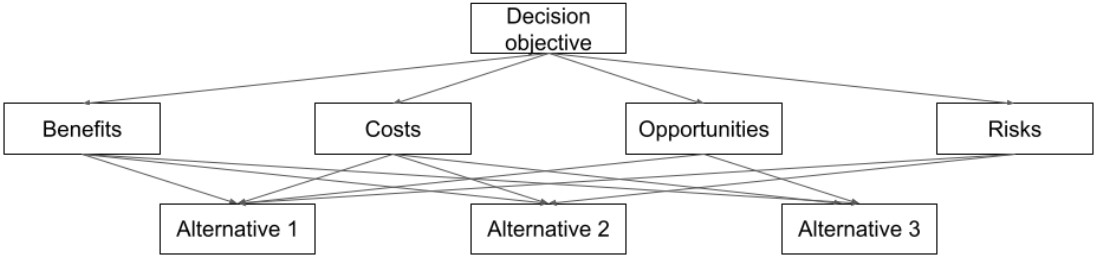

**Figure 2.** Three-level BOCR model with an incomplete hierarchy.

The incomplete situation presented in Figure 2 is not clear, since it just involves the removal of two arrows from Figure 1. Therefore, an organizational chart, such as the one presented in Figure 3, would be preferred, being clearer in showing the incompleteness.

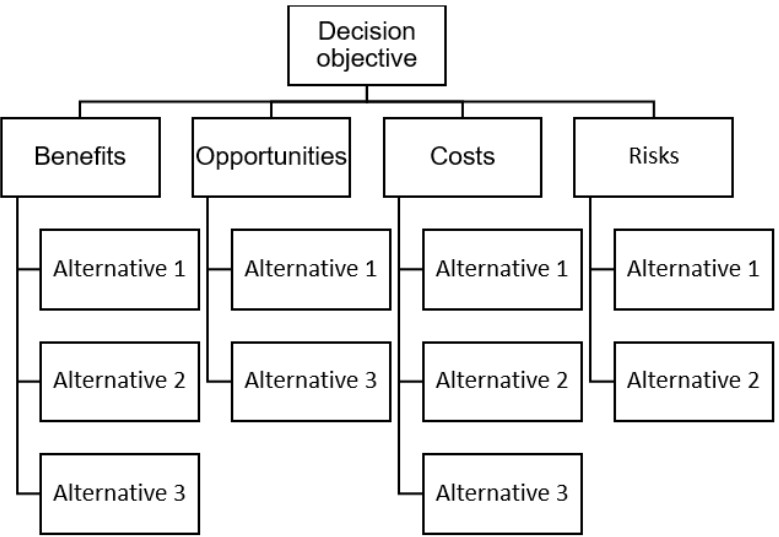

**Figure 3.** Organizational chart for a three-level BOCR model with an incomplete hierarchy.

In the decision problem presented in Figures 2 and 3, alternative two is not assessed according to opportunities and alternative three is not accessed according to risks. One possible explanation is that there is no positive consequence for alternative two. Therefore, there will be no opportunity gained from choosing it. Alternatives one and three, in contrast, promote future benefits. However, if alternative three is not evaluated according to its risks, there are two possible explanations. First, this alternative may be too risky compared to alternatives one and two. Therefore, alternative three's priority in terms of risks would be equal to zero. Another possibility is the opposite: alternatives one and two may be too risky compared to alternative three. In that case, alternative three's priority will also be zero, but the priorities of alternatives two and three will be negative (Saaty and Ozdemir 2003).

Therefore, as previously discussed, the BOCR model is not only useful for the structuring step. The use of the model also impacts the steps of measuring and synthesis. In measuring, it is important to clearly express how the costs and risks are measured. Their priorities may be positive or negative. In the AHP, priorities are obtained with the right eigenvector $w = [w_i]$ of the comparison matrix $A = [a_{ij}]$. If an alternative $i$ is more costly than another $j$, and $a_{ij} < 1$, there are positive priorities. The less costly alternative receives the greatest $w_i$. However, if an alternative $i$ is more costly than $j$, and $a_{ij} > 1$, there are negative priorities: the most costly alternative receives the greatest $w_i$. Interestingly, negative priorities do not imply negative numbers. As a matter of fact, with $w$ obtained from $A$, $w_i > 0$ for $i = 1, 2, 3 \ldots n$, where $n$ is the number of alternatives.

For incomplete hierarchies, as previously discussed, it is possible that $w_i = 0$. If the priorities are not obtained through pairwise comparison—for instance, by using direct data—it is possible that $w_i < 0$. Therefore, there are four ways to combine the alternatives' priorities according to each criterion ($b, o, c$, and $r$), resulting in the overall priority $x$, as in Equations (1)–(4), where $B, O, C$, and $R$ are the weights of each criterion.

$$x = Bb + Oo + Cc + Rr \tag{1}$$

$$x = Bb + Oo + C(1/c) + R(1/r) \tag{2}$$

$$x = Bb + Oo + C(1 - c) + R(1 - r) \tag{3}$$

$$x = \frac{b^B o^O}{c^C r^R} \tag{4}$$

Only in Equation (1) are there positive priorities. Equation (2) has negative priorities obtained with comparison matrices. Equation (3) deals with the concept of residual or complimentary values. Equation (4) highlights a tradeoff between benefits and costs and

costs and risks. These different equations express the impacts of the use of the BOCR model in the synthesis step.

The BOCR model has also been used in a combined manner in several approaches, such as the AHP–TOPSIS (Hsueh and Lin 2017; Wei 2021), ANP–TOPSIS (Liu and Yin 2019), DEMATEL (Lewis Barrios et al. 2013; Tsai and Chen 1991), fuzzy sets (Charoennapharat and Chaopaisarn 2021), fuzzy AHP (Alberto and Julian 2016; Chen et al. 2017; Ho and Hsu 2020; Hsueh and Lin 2017; Ilbahar et al. 2022; Kim and Yang 2023), fuzzy ANP (Karabece and Gurbuz 2020; Sadeghi and Larimian 2018), fuzzy and grey theories (Liu et al. 2020), prospect theory and Choquet integral (Qian et al. 2021), TODIM (Qian et al. 2019), and VIKOR (Hadian et al. 2020; Tsai and Lin 2018) approaches.

The BOCR approach has been applied in various fields, such as business, economics, engineering, management, and the social sciences. The work by Saaty (2013) highlighted some projects conducted with the BOCR approach:

- Determining whether advertising agency clients should be kept;
- Student admissions;
- Determining whether affirmative action is still needed;
- Allocating resources to a department;
- Baseball: Diamond vs. Indians;
- Determining the best career decisions;
- Determining the best smoking policy;
- Bids for an engineering project;
- Casual corner apparel strategic business decisions;
- Choice of the vendor for new software;
- Choosing the best alternative to reduce the effects of gasoline price volatility for automobiles in the United States;
- A credit card expansion strategy;
- A decision model for company downsizing;
- Entrepreneurial development decisions;
- A food model;
- Green Bay Packers' best allocation of resources under a salary cap;
- Human tracking system decisions;
- A management issues project;
- Outsourcing IT;
- Project resource allocation.

## 3. Methodology

There are a dozen different approaches to literature reviews (Grant and Booth 2009), as presented in Figure 4.

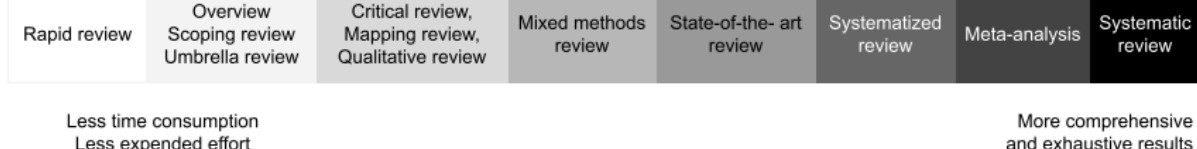

**Figure 4.** Spectrum of literature review approaches (Ruiz Bargueno et al. 2021).

To meet the objective proposed in Section 1, a state-of-the-art review was conducted. As presented in Figure 1, results from a state-of-the-art review can be expected to be more comprehensive than those from a rapid review, umbrella review, or qualitative review of the literature (Ruiz Bargueno et al. 2021). However, results from a state-of-the-art review should not be expected to be as exhaustive as those obtained with a systematic review of the literature. Since a state-of-the-art review aims to review more current work, the time span does not need to cover long periods, such as decades (Higgins and Thomas 2016).

This review is based on searches performed in the Scopus database, a widely recommended database for literature reviews (Burnham 2006; Oliveira et al. 2018). Most literature reviews search Clarivate's Web of Science or Elsevier's Scopus databases (Mongeon and Paul-Hus 2016). However, both databases have similar contents. Therefore, the Scopus database was chosen for the state-of-the-art review to ensure the uniformity of data, search strings, and results and also because of the previous literature review successfully performed by the authors with this database (Martino Neto et al. 2020; Oliveira et al. 2022; Ruiz Bargueno et al. 2021; Tramarico et al. 2015).

The state-of-the-art review followed the four-step method proposed by Ruiz Bargueno et al. (2021), with one adaptation in step three, as presented in Figure 5.

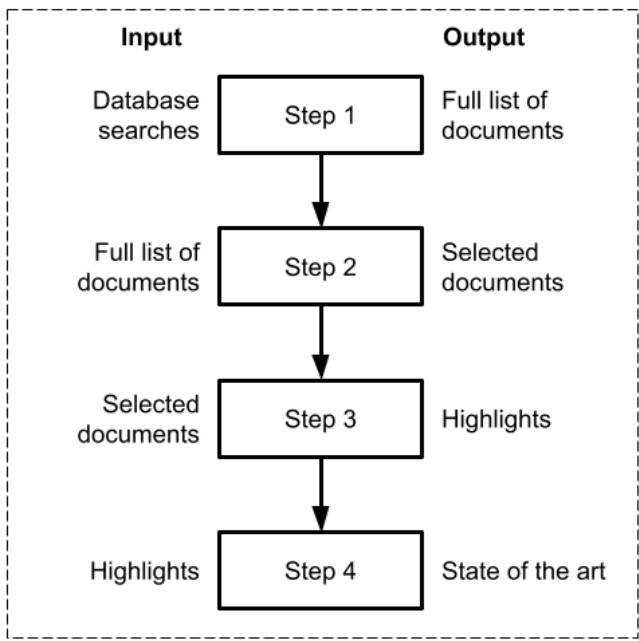

**Figure 5.** Four-step state-of-the-art review method (adapted from Ruiz Bargueno et al. (2021)).

In step one, the Scopus database (www.scopus.com (accessed on 21 April 2023)) was searched in April of 2023. As discussed in Section 4, some documents employing the term "BOCR" were not on benefits, opportunities, costs, and risks analysis. This was due to the use of the string "BOCR" as a variable; for instance, $B_{0_{CR}}$ in Hassan and Wang (2019). Therefore, the number of documents was reduced in step two.

For step three, Ruiz Bargueno et al. (2021) performed a keyword analysis within a network analysis implemented in the software VOSViewer (www.vosviewer.com (accessed from January to June 2021)). The review presented in this paper goes further, presenting highlights concerning years of publication (Figure 6), sources (Table 1), areas (Figure 7), and countries (Figure 8). No complimentary software was necessary, making the method more efficient. Here, the four-step method was performed only with the figures provided by Scopus.

After highlighting the most cited documents on BOCR (Table 2), Section 6 summarizes the state-of-the-art for AHP methods with BOCR. Then, despite minimal use of other theories, it is proven that the acronym BOCR mostly belongs to the AHP theory.

**Table 1.** Documents on BOCR by source.

| Source | Publisher | Documents |
|---|---|---|
| *Sustainability* | MDPI | 7 |
| *Int. J. of Logistics Systems and Management* | Inderscience | |
| *Int. J. of Production Research* | Taylor and Francis | |
| *Int. J. of the Analytic Hierarchy Process* | Creative Decisions | 4 |
| *J. of Testing and Evaluation* | ASTM | |
| *Advances in Intelligent Systems and Computing* | Springer Nature | |
| *IFAC-PapersOnLine* | Conference proceedings | |
| *Int. J. of Information Technology and DM* | World Scientific | 3 |
| *J. of Cleaner Production* | Elsevier | |
| *Chemical Eng. Transactions* | AIDIC | |
| *Clean Technologies and Environmental Policy* | Springer Nature | |
| *Cogent Eng.* | Cogent | |
| *Computers and Industrial Eng.* | Elsevier | |
| *Energy Conversion and Management* | Elsevier | |
| *Expert Systems with Applications* | Elsevier | |
| *Gospodarka Surowcami Mineralnymi* | Polish Academy of Sciences | |
| *ICPS Technical Conference* | Conference proceedings | |
| *IEEE Int. Conference on SOLI* | Conference proceedings | |
| *Int. J. of Advanced Manufacturing Technology* | Springer Nature | |
| *Int. J. of Green Energy* | Taylor and Francis | 2 |
| *Int. J. of Production Economics* | Elsevier | |
| *Int. J. of System Assurance Eng. and Management* | Springer Nature | |
| *J. of Forestry Research* | Northeast Forestry University | |
| *Lecture Notes in Computer Science* | Springer Nature | |
| *MATEC Web of Conferences* | Conference proceedings | |
| *Omega* | Elsevier | |
| *PLOS One* | PLOS | |
| *Proceedings of SPIE* | Conference proceedings | |
| *Renewable and Sustainable Energy Reviews* | Elsevier | |
| *Resources Policy* | Elsevier | |
| *Technological and Economic Dev. of Economy* | VILNIUS TECH | |

**Table 2.** Most cited documents on BOCR.

| | Document | Citations | Subject |
|---|---|---|---|
| 1 | Demirtas and Ustun (2008) | 310 | Supply chain management |
| 2 | Lee (2009) | 305 | Supply chain management |
| 3 | Lee et al. (2009) | 232 | Energy and supply chain management |
| 4 | Ustun and Demirtas (2008) | 159 | Supply chain management |
| 5 | Kabak and Dagdeviren (2014) | 156 | Energy management |
| 6 | Demirtas and Ustun (2009) | 146 | Supply chain management |
| 7 | Alizadeh et al. (2020) | 134 | Energy management |
| 8 | Wijnmalen (2007) | 119 | Methodology |
| 9 | Yap and Nixon (2015) | 116 | Project management |
| 10 | Liang and Li (2008) | 97 | Information technology |

## 4. Results

Initial searches with title–abstract–keyword (TITLE-ABS-KEY) in the Scopus database resulted in the data shown in Table 3.

**Table 3.** Initial searches in Scopus.

| TITLE-ABS-KEY | Documents | Top Reference by Citations |
|---|---|---|
| "ahp" or "analytic hierarchy process" | 50,388 | First: Saaty (1990), 5825 |
| "bocr" | 191 | First: Demirtas and Ustun (2008), 310 |
| "mcdm" or "multi-criteria" or "multiple" and "criteria" | 182,051 | Tenth: Saaty (1977), 6155 |

Initially, there were more documents in the Scopus database on MCDM than on the AHP, and more documents on the AHP than on BOCR. However, the most cited documents for the keywords "multi-criteria" were not related to MCDM but to medicine and pharmacy studies. For instance, the most cited document (Spitzer et al. 2006) did not have MCDM in the TITLE-ABS-KEY. As a matter of fact, it was a paper on anxiety disorder that did not address MCDM methods and concepts. There, the word "criterion" was not used for the concepts of an "attribute" or "objective" but for the concept of a "rule". The next eight most cited documents relating to "multi-criteria" or "multiple criteria" were also not on MCDM. They were from the fields of biology, medical care, and neurology. The top document actually on MCDM was a seminal paper on the AHP (Saaty 1977). It is interesting that it was not found in the search for AHP documents because that paper preceded the proposal of the name AHP (Saaty 1980).

Table 4 presents results of searches for only the acronyms AHP, BOCR, and MCDM, as well as MCDA, which may stand for "multiple criteria decision aid" or "multiple criteria decision analysis". All three top references ranked first in their search results.

**Table 4.** Acronym-restricted searches in Scopus.

| TITLE-ABS-KEY | Documents | Top Reference by Citations |
|---|---|---|
| "ahp" | 39,720 | Chang (1996), 3300 |
| "bocr" | 191 | Demirtas and Ustun (2008), 310 |
| "mcda" or "mcdm" | 16,943 | Opricovic and Tzeng (2004), 2971 |

The search using the TITLE–ABS–KEY ("ahp" AND "bocr") in July 2023 resulted in 63 documents, as reported by reviewer one. It is interesting to note that these were not all documents on the AHP and BOCR. For instance, there were documents without AHP in the title, abstract, or keywords (Alizadeh et al. 2020; Demirtas and Ustun 2008; Kabak and Dagdeviren 2014; Liang and Li 2008; Ustun and Demirtas 2008). For that reason, these most cited documents were not listed in that search.

With the acronym restriction, the order MCDM–AHP–BOCR was reversed to AHP–MCDM–BOCR. This means that the acronym AHP had more publications than the acronym MCDM. Thus, the leadership of AHP among MCDM methods could be confirmed. However, in 16 of the top 100 documents, AHP meant not "analytic hierarchy process" but "adult hippocampal progenitors" (Mody et al. 1987; Suhonen et al. 1996), "afterhyperpolarization" (Anwyl 1999; Horvath et al. 1999; Hotson and Prince 1980; Llinas and Yarom 1981; Marrion and Tavalin 1998; McIntyre et al. 2002; Nicoll et al. 1990; Rudy 1988; Sah 1996; Steriade et al. 1993; Storm 1990; Traub et al. 1991), "alkyl hydroperoxide reductase" (Seaver and Imlay 2001), or "atomic hybrid polarizabilities" (Miller 1990). With this proportion, there would be 33,364 documents on the AHP method, still more than twice the number of documents published on MCDM.

Only 6 of the 20 most cited documents on MCDM (Chai et al. 2013; Huang et al. 2011; Kumar et al. 2017; Malczewski 2006; Pohekar and Ramachandran 2004; Wang et al. 2009) did not have the AHP acronym in the TITLE-ABS-KEY. This result confirmed the

leadership of the AHP among the various MCDM methods applied in energy management, environmental sciences, geographical information, and supply chain management.

In the 15th most cited document, BOCR stood for "bifocal optical coherence refractometry" (Zvyagin et al. 2003). The 50th document, on "biological contact oxidation reactor" (BCOR), included a mistyped BOCR in its abstract. The 56th and the 73rd documents included the critical magnetic bond number $B_{O_{CR}}$ in their abstracts (Hassan and Wang 2019, 2020). The 57th and the 116th documents included other variables: the Remanence coercivity $B_{OCR}$ (Kumaravel et al. 2005) and the buoyancy critical value $Bo_{cr}$ (Zujus et al. 2009). In the 117th document, BOCR stood for "blockchain and online customer review" (Wan et al. 2022), and in the 161st, it stood for "biological oxygen consumption rate" (Eguchi et al. 2002). Finally, in two documents that had not been cited, BOCR stood for "Bayesian optimization with clustering and rollback" (Fan et al. 2022) and "bifocal optical coherence refractometry" (Alexandrov et al. 2005). Thus, 181 of the 191 documents published referencing BOCR were on "benefits, opportunities, costs, and risks". This rate of occurrence greater than 94% implies that the acronym BOCR belongs to the MCDM methodology; specifically, the AHP and ANP methods.

Figure 6 presents the numbers of documents where TITLE-ASB-KEY = "bocr" year by year from 2002 (blue curve). The figure also presents the accumulated documents year by year (green curve). This second set of information is important to evaluate the maturity of the subject under study in terms of publications.

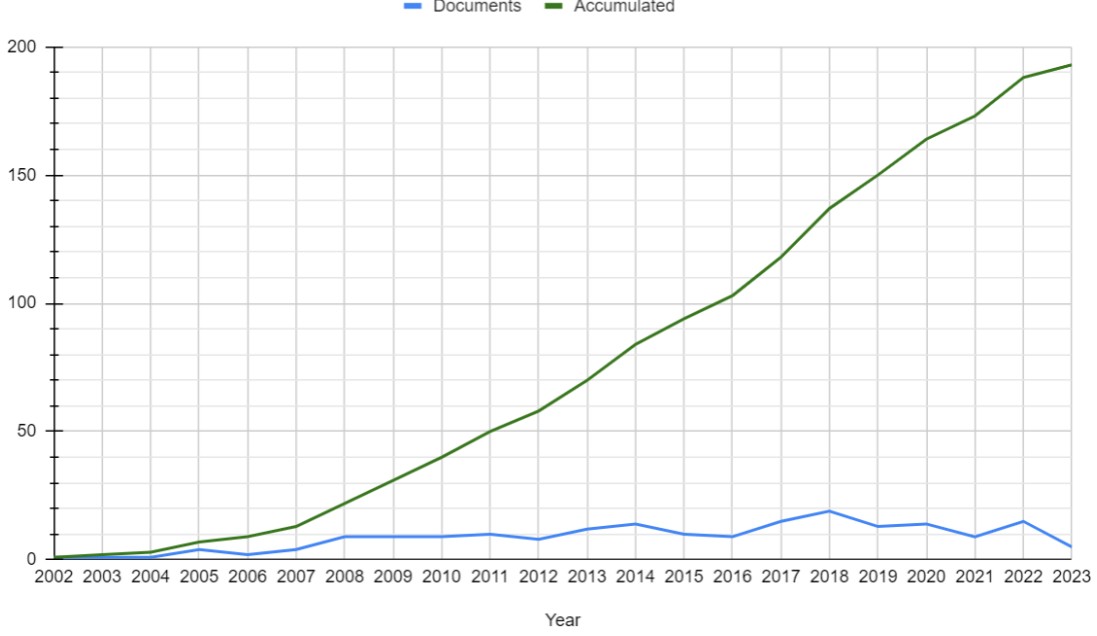

**Figure 6.** Documents on BOCR by year and accumulated number of documents.

The publication of documents on BOCR grew year by year until 2018 when it reached a peak of 19 published documents. After this year, the publication rate remained below 15 documents per year, with only 9 documents indexed in 2021. However, as the accumulated curve has not reached an inflection point, this indicates that this field of knowledge has not reached maturity or that the interest of the scientific community has not cooled down (Price 2020). The decrease in annual publications suggests that BOCR is a topic that needs to be explored by scholars researching the AHP method. After all, as the subject is not out of interest, this small decrease indicates a demand for publications on this subject.

Table 1 presents the main sources of BOCR publications. In addition to the 31 sources listed in the table, a further 112 sources have published only one document on BOCR. Documents are well distributed across the 143 different sources, averaging 1.33 documents/source. Therefore, there is no saturation of BOCR research in any specific publica-

tion source. Rather, BOCR is a promising theme for publication by several publishers, such as Elsevier, MDPI, Springer Nature, and Taylor and Francis, to name a few.

Figure 7 presents the distribution of BOCR documents published by subject area. The top areas are engineering, computer science, business, decision sciences, energy, environmental sciences, and mathematics. Together, decision sciences and mathematics represent less than 13.5% of publications, suggesting that documents on BOCR are extremely practical.

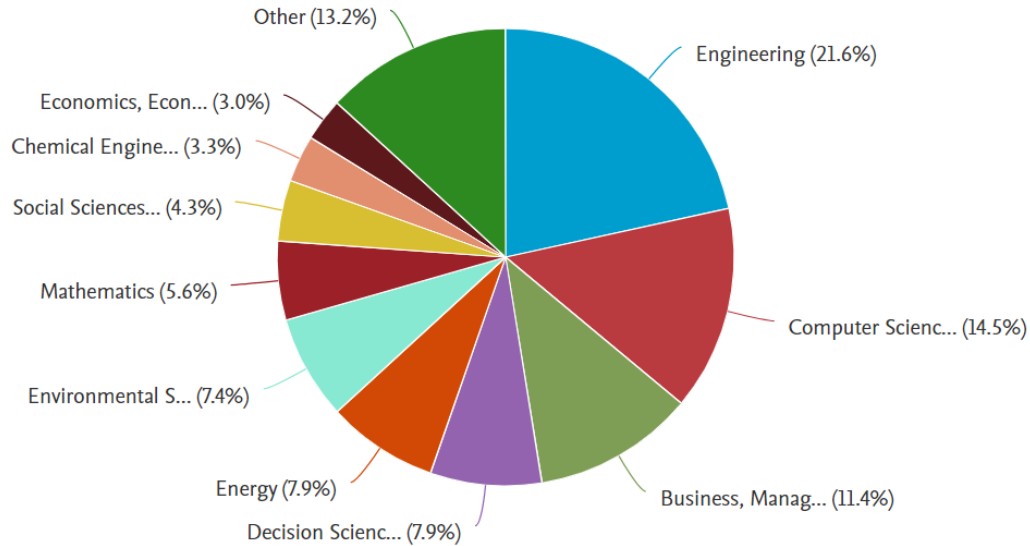

**Figure 7.** Documents on BOCR by subject area (source: www.scopus.com (accessed on 21 April 2023)).

Figure 8 presents the top ten countries where the first authors of documents are affiliated. Scopus counts China, Hong Kong, Macao, and Taiwan independently. Together, they have 63 published documents or almost a third of BOCR publications. Separately, Taiwan and China lead in authorship, with 34 and 24 authored documents, respectively.

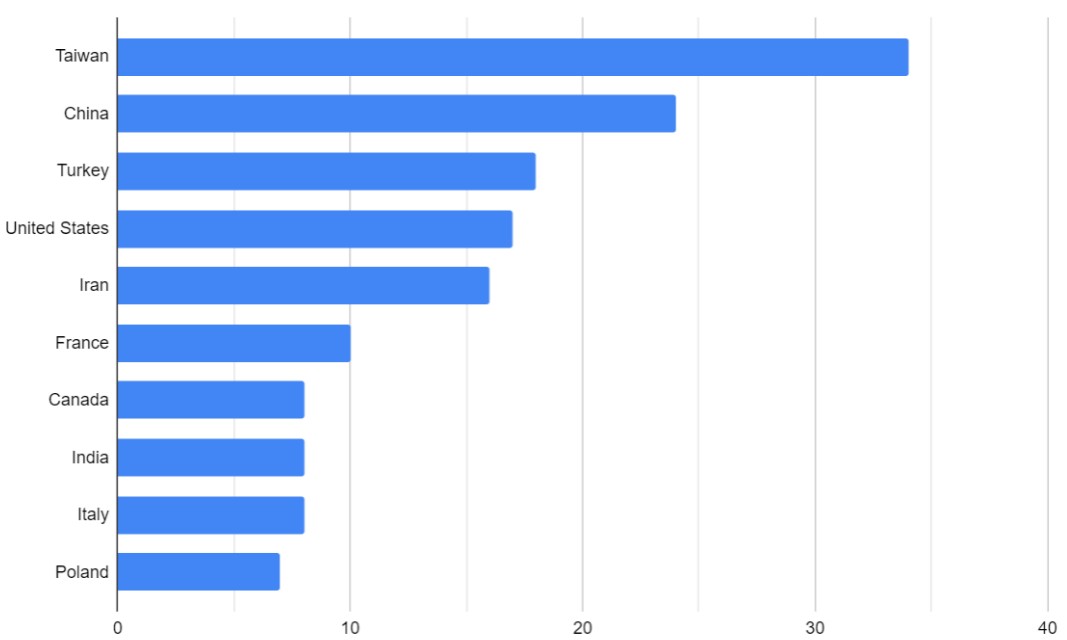

**Figure 8.** Documents on BOCR by country.

## 5. Discussion

Table 2 presents the top 10 most cited documents on BOCR.

Amy H.I Lee, from Taiwan, is the most prolific author, with 15 published documents on BOCR. However, the three documents authored by Ezgi Aktar Demirtas and Ozden Ustun, from Turkey, have higher numbers of citations.

Energy management and supply chain management were the most researched subjects in the BOCR documents. This does not imply that BOCR research on those subjects is exhausted. However, it indicates that research on other practical subjects, such as education, government, healthcare, industrial and manufacturing engineering, innovation and entrepreneurship, public policy, social responsibility, and sustainability, would be welcome (https://isahp.org/callforpapers). Methodological contributions would also be welcome, since there was only one document among the top 10 on the BOCR methodology (Wijnmalen 2007).

To summarize, there were three main findings from the literature review. First was that the BOCR acronym belongs to the MCDM methodology, with 94% of documents published being on the MCDM methodology, specifically the AHP or ANP methods. The second was the distribution of the 191 documents found, which were published in 143 different sources, with MDPI's *Sustainability* leading with 7 documents published. The diversity of sources was observed from the existence of several publishers. The last main result was that most of the publications were practical documents on business, computer sciences, and engineering.

## 6. Conclusions

The BOCR model is an MCDM model employed to elicit a mutually exclusive and collectively exhaustive set of criteria. As an acronym proposed in AHP theory, it has received attention from the users of this method. This paper intended to answer the question: how have AHP and BOCR been applied? To answer that question, the paper presented the state-of-the-art of the AHP and BOCR literature. Documents found in the Scopus database mostly concerned practical applications but covered several fields, including business, computer science, and engineering. Specific objectives were to identify research gaps and propose future research themes. Based on the findings presented in Sections 5 and 6, the main areas for future research on BOCR should be:

- Methodological contributions—for instance, comparing the BOCR model with other models, such as environmental, social, and corporate governance (ESG); strengths–weaknesses–opportunities–threats (SWOT) analysis; and value, rarity, imitation, and organization (VRIO) questions (Barney and Hesterly 2005);
- Practical contributions applying BOCR in other areas different from the usual fields of business, computer sciences, and engineering.

**Author Contributions:** Conceptualization, A.P. and V.A.P.S.; methodology, A.P. and V.A.P.S.; software, A.P. and V.A.P.S.; validation, A.P., C.L.T., and V.A.P.S.; formal analysis, A.P., C.L.T., and V.A.P.S.; investigation, A.P., C.L.T., and V.A.P.S.; resources, A.P., C.L.T., and V.A.P.S.; data curation, A.P., C.L.T., and V.A.P.S.; writing—original draft preparation, A.P., C.L.T., and V.A.P.S.; writing—review and editing, A.P., C.L.T., and V.A.P.S.; visualization, A.P., C.L.T., and V.A.P.S.; supervision, A.P. and V.A.P.S.; project administration, A.P. and V.A.P.S.; funding acquisition, A.P. and V.A.P.S. All authors have read and agreed to the published version of the manuscript.

**Funding:** This research received no external funding.

**Data Availability Statement:** The researched database is available at http://www.scopus.com (accessed on 21 April 2023).

**Acknowledgments:** The authors must thank the anonymous reviewers for their valuable comments, criticisms, and suggestions. Specifically, reviewer one indicated that the TITLE–ABS–KEY search ("ahp" AND "bocr") resulted in 63 documents, as noted in Section 4.

**Conflicts of Interest:** The authors declare no conflict of interest.

## Abbreviations

The following abbreviations are used in this manuscript:

| | |
|---|---|
| AIDIC | Associazione Italiana Di Ingegneria Chimica |
| AHP | Analytic Hierarchy Process |
| ANP | Analytic Network Process |
| ARAS | Additive Ratio Assessment |
| ASTM | American Society for Testing and Materials |
| BCA | benefit–cost analysis |
| BOCR | benefits, opportunities, costs, and risks |
| COPRAS | Complex Proportional Assessment |
| DEMATEL | Decision-Making Trial and Evaluation Laboratory |
| Dev. | development |
| DM | decision making |
| ELECTRE | Elimination et Choix Traduisant la Realite |
| Eng. | engineering |
| ESG | environmental, social, and corporate governance |
| GP | Goal Programming |
| ICPS | industrial and commercial power systems |
| IEEE | Institute of Electrical and Electronics Engineers |
| Int. J. | international journal |
| MACBETH | Measuring Attractiveness by a Categorical-based Evaluation Technique |
| MATEC | materials science, engineering, and chemistry |
| MAUT | Multi-Attribute Utility Theory |
| MCDA | multiple-criteria decision aid |
| | multiple-criteria decision analysis |
| MCDM | multiple-criteria decision making |
| MDPI | Multidisciplinary Digital Publishing Institute |
| MECE | mutually exclusive and collectively exhaustive |
| PLOS | Public Library of Science |
| PROMETHEE | Preference Ranking Organization Method for Enrichment of Evaluations |
| SOLI | service operations, logistics, and informatics |
| SWOT | strengths–weaknesses–opportunities–threats |
| TOPSIS | Technique for Order Preference by Similarity to Ideal Solution |
| VIKOR | Visekriterijumska Optimizacija i Kompromisno Resenje |
| VILNIUS TECH | Vilnius Gediminas Technical University |
| VRIO | value, rarity, imitation, and organization |

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
