# Peer review of "State-of-the-Art Review on the Analytic Hierarchy Process with Benefits, Opportunities, Costs, and Risks"

_jrfm, doi:10.3390/jrfm16080372_

Round 1
Reviewer 1 Report
1. The title of the article was treated superficially. Simple queries related to the topic were used.
2. Subsection 2.1. is more suited to an introduction than a methodology.
3. Subsection 2.2. The methodology used in the research should be developed and presented in more detail.
4. From the point of view of the title, the results for a simple query are missing: “( TITLE-ABS-KEY ( bocr ) AND TITLE-ABS-KEY ( ahp ) )”. These 63 (2023-07-20) documents (“PUBYEAR > 2001 AND PUBYEAR < 2024“) should be analyzed.
5. The conclusions from the work are not clear in the light of the goals set.
6. (line 190 – 194) – This rather concerns Figure 4.
7. “The decrease in annual publications suggests BOCR is a topic to be explored by researchers of the AHP method.” (line 249-250). The conclusion should be explained more precisely and logically.
8. The figures are simply and directly copied from the query response. Figure 7. Description of the x axis - the numbers on the axis should be integers. For example for “( TITLE-ABS-KEY ( bocr ) AND TITLE-ABS-KEY ( ahp ) ) AND PUBYEAR > 2001 AND PUBYEAR < 2024” :
Author Response
Point 1: "Literature review" replaced with "State of the Art Review" in the title to clarify that this manuscript does not deliver a Systematic Review of the Literature.
Point 2: Subsection 2.1 changed to Section 2 Background, since BOCR is not a method adopted in this manuscript.
Point 3: Subsection 2.2 changed to Section 3 Methodology
Point 4: The search TITLE–ABS–KEY(“ahp” AND “bocr”) resulting in 63 documents was commented after Table 2 and the Reviewer 1 was thanked in the Acknowledgments. Documents were not analyzed because this is a state-of-the-art review, not a systematic review.
Point 5: Two lines were added in Conclusions linking them with the objectives of the paper.
Point 6: Original Lines 190-194 (now Lines 211-215) called and commented Table 1. We are very sorry, but we cannot understand what is the concern with Fig. 4.
Point 7: The conclusion is based on the fact that BOCR publications is still attracting attention, or is not cooled down (as commented in Price, 2020). So, we updated Fig. 6 and added some lines and the new reference on that.
Point 8: A new Fig. 7 was uploaded with integers in x-axis. This figure was generated by Google Sheets, not by the Scopus analysis tool.
Reviewer 2 Report
The language used is convoluted and difficult to understand, making it challenging for readers to grasp the intended message.
The author makes sweeping claims about the prominence of BOCR in the MCDM method users without providing concrete evidence or references. This weakens the credibility of the statements presented.
The term "State of the Art Review" is used without adequate explanation or context, leaving readers puzzled about its significance in the research.
While the passage mentions diverse areas where BOCR has been applied, it fails to elaborate on specific examples or case studies, making it difficult to assess the real-world impact and effectiveness of the model.
The description of "Methodological and Practical" proposals for future research on BOCR lacks specificity. It would be more informative to include concrete suggestions or areas where further investigation is needed.
There is a lack of critical analysis regarding the limitations or weaknesses of the BOCR model. Every decision-making approach has drawbacks, and ignoring these aspects undermines the overall credibility of the discussion.
The reference to the Scopus database as the source of the literature review raises questions about the comprehensiveness and reliability of the study. The absence of more details regarding the search criteria and methodology is concerning.
The passage does not mention any potential alternatives to the BOCR model or compare its effectiveness to other decision-making approaches, leaving readers with an incomplete understanding of its value.
It is unclear what the author means by a "mutually exclusive and collectively exhaustive set of criteria" in the context of the BOCR model. Further elaboration is needed to make this concept understandable to the readers.
The paper lacks a strong conclusion summarizing the key findings and implications of the literature review.
Should be improved.
Author Response
We accept the criticism of the use of language, even because none of the authors are native speakers of English. In this version we use a language revision service and promote several corrections, all highlighted with red text.
From the title, the main change was the identification that this paper delivers a state-of-the-art review, that's all. Regarding the use of the expressions ''state-of-the-art'' and ''state of the art'', this version distinguishes between state-of-the-art as the quality of the review, and state of the art as an object, or product to be delivered. In the section Methodology, it is explained that the state-of-the-art review proposes to be more elaborate than a quick review, but less exhaustive than a systematic review.
The comment "The description of "Methodological and Practical" proposals for future research on BOCR lacks specificity. It would be more informative to include concrete suggestions or areas where further investigation is needed." was already answered with two bullet points in the section Conclusions.
On the use of Scopus, new lines were added to address this comment, including new references (Martino et al. 2020, Mongeon and Paul-Hus. 2016, Oliveira et al. 2022).
On the "mutually exclusive and collectively exhaustive set of criteria" a new text including a reference from Oxford University was added in the Introduction.
Round 2
Reviewer 1 Report
ad. 8. Please check Fig. 8 (prev. 7), numbers are still non-integer. It looks like directly copied from the query response.
https://www.scopus.com/term/analyzer.uri?sort=plf-f&src=s&sid=a0d9173b1ad1e2672a303d9b705ecade&sot=a&sdt=a&sl=19&s=TITLE-ABS-KEY%28bocr%29&origin=resultslist&count=10&analyzeResults=Analyze+results
Author Response
The current Fig. 8 (generated by the authors in Google Sheets) replaced the original Fig. 7 (generated by Scopus):

Please, note that the new figure has only integer numbers in x-axis. We cannot find how to change the settings of previous figure in Scopus. Also, there were no country with a non-integer publication number. It was only the scale factor of the x-axis automatically set by Scopus.